# Estimation of Particle Location in Granular Materials Based on Graph Neural Networks

**DOI:** 10.3390/mi14040714

**Published:** 2023-03-23

**Authors:** Hang Zhang, Xingqiao Li, Zirui Li, Duan Huang, Ling Zhang

**Affiliations:** 1School of Automation, Central South University, Changsha 410083, China; 2School of Computer Science and Engineering, Central South University, Changsha 410083, China

**Keywords:** particle locations, graph convolutional network, coordinate prediction, two-dimensional photoelastic granular materials, distance information, distance estimation algorithm

## Abstract

Particle locations determine the whole structure of a granular system, which is crucial to understanding various anomalous behaviors in glasses and amorphous solids. How to accurately determine the coordinates of each particle in such materials within a short time has always been a challenge. In this paper, we use an improved graph convolutional neural network to estimate the particle locations in two-dimensional photoelastic granular materials purely from the knowledge of the distances for each particle, which can be estimated in advance via a distance estimation algorithm. The robustness and effectiveness of our model are verified by testing other granular systems with different disorder degrees, as well as systems with different configurations. In this study, we attempt to provide a new route to the structural information of granular systems irrelevant to dimensionality, compositions, or other material properties.

## 1. Introduction

Granular materials are composed of a large number of particles, which can show both liquid-like disordered structures and solid-like stable mechanics [1,2,3]. Accordingly, granular materials can be used as a simplified glass model to understand various complex physical behaviors of glasses and amorphous solids, such as glass transition [4,5], jamming transition [6,7], vibrational modes [8,9], plasticity [10,11], earthquake dynamics [12], etc., which are mostly based on the structure of granular systems. Additionally, the structural analysis of granular materials is especially critical for engineering applications, industrial production [13], or even geological disasters such as avalanches and mudslides [14,15,16]. Therefore, visualization of the structures in granular materials plays an important role in the understanding of anomalous behaviors in glasses and amorphous solids and the prevention of geological disasters. However, due to the limitation of experimental techniques, the acquisition of the inner structures of granular materials has been a great challenge and also a research focus for many scientists.

One of the simplest methods to obtain the structure of granular materials is naturally to use a camera to directly photograph it. However, for a three-dimensional (3D) granular system, this method cannot obtain its internal structure, and when analyzing the structure, this method is also limited by the resolution of the camera. On the basis of scattering, diffraction, and electron imaging technology, researchers have tried to use neutron scattering [17], X-ray [18], electron microscopy [19], sound wave diffraction [20], and other methods to analyze the structure of granular systems. Although using these techniques can derive the overall structure of the granular system from the spectral analysis, one cannot accurately obtain grain-scale micro-structural information. More importantly, expensive equipment substantially increases the experimental costs and difficulties. Speckle visible spectroscopy [21,22] has also been used to analyze the structure of granular systems. The principle of this technique is that when the laser diffuses on the surface of the scatterer or passes through a transparent scatterer, an irregular distribution of bright and dark spots can be observed in the light field on or near the scattering surface. We can then derive the structure of the scatterer from the spectral analysis. Similarly, this method cannot accurately obtain grain-scale micro-structural information. Recently, researchers have used computational imaging technology [23] to reconstruct the structure of blocked objects. However, this method requires that the shelter must be stationary, the hidden object radiates or scatters coherent light, and its motion information must be known. To compensate for the deficiencies of these techniques, in this paper, we introduce an improved graph convolutional neural network (GCN) to achieve the grain-scale micro-structures of granular materials. Compared with the classical GCN model, the GCN model used in this paper adds a hidden representation in the feature propagation to mark the hidden representation of nodes. In addition, we introduce a hop threshold to the GCN model as the basis for constructing the adjacency matrix. We provide a detailed description of the hidden representation and the hop threshold in Section 4.

In recent years, graphs as effective data structures have been used in various fields. The nodes in the graph often contain a lot of valuable data, which are often not handled properly with traditional machine learning models (including deep learning models) [24,25]. The advantage of a graph neural network (GNN) is that it can represent the information near nodes at any depth and use this information in an appropriate way. This advantage enables GNNs to handle various graph-related learning tasks well. Consequently, GNNs have achieved many state-of-the-art results in various graph-related learning tasks, such as node classification, link prediction, and graph classification [26,27,28,29]. For example, in the field of biological science [30], GNNs have achieved protein structure prediction [31,32,33], drug-related prediction [34], disease prediction [35,36], biomedical imaging [37], etc. Similarly, GNNs have been introduced into the field of particle physics [38,39] and have made remarkable achievements, such as glassy dynamics prediction [40,41], crystal structure prediction [42,43,44,45], and force chain structure prediction [46,47]. Although GNNs have successfully predicted the above problems, it has always been a huge challenge to predict the structure of granular systems. Because of the complexity of granular systems, GNNs need to extract more information to predict the structure of the granular system. However, the core issue of the structure of a granular system is the grain-scale position. When the position of each particle in the granular system is known, the structure of the granular system can be directly constructed.

When locating a particle in a granular system, we need to measure the coordinates of some particles (anchor nodes), and also need to know the distance and angle between the anchor node and the unknown node. However, in this paper, we can use the GNN model to predict the coordinates of unknown nodes when we know the distance information between anchor nodes and unknown nodes [48]. Thus, we need to aggregate the distance information of the whole granular system as the input of the GNN. In granular systems, the distance between two non-adjacent particles is often difficult to measure. Consequently, we need to introduce a distance estimation algorithm to estimate the distance between two non-adjacent particles. Distance estimation algorithms can be divided into two categories depending on whether additional equipment is required for measurement: range-based and range-free. Range-based algorithms are implemented by using extra hardware devices to measure some range parameters such as time of arrival (TOA) [49], time difference of arrival (TDOA) [50], angle of arrival (AOA) [51], etc. Range-free algorithms are directly based on distance estimation between nodes, such as distance vector hop (DV-Hop) [52], approximate point in triangle (APIT) [53], convex position estimation (CPE) [54], and so on. A range-free algorithm combines its own information with the relevant information of neighbors to estimate the distance between two nodes without the need for additional equipment for measurement. Relative to a range-based algorithm, the accuracy of a range-free algorithm is lower. However, the distance between particles is generally unmeasurable in this project, as we based our study on an improved range-free algorithm [55] to estimate the distance between particles in the granular system.

In this paper, we propose a GCN-based method for calculating particle coordinates in two-dimensional (2D) photoelastic granular systems, based on the estimated distance information. Specifically, we first generate the data sets for the GNN model by performing the photoelastic experiment of 2D granular systems with different disorder degrees and configurations. Then, we use the distance estimation algorithm to estimate the distance information for each particle in all granular systems. To train the GCN model, we take the distance information of a granular system as the input and output coordinates of all particles in the system. We use the nodes from about 60% of the entire particle system as the training set to train the GCN model. We also define the nodes in the training set as anchor nodes, in which the coordinates of nodes are known. In order to comprehensively evaluate the model, we use the root-mean-squared error (RMSE) to evaluate the stability and robustness of the model, and the effective prediction accuracy (EPA) to evaluate the prediction accuracy of the model. We then test the model in some granular systems, and we achieve a small RMSE and a high EPA. Finally, to verify the effectiveness of our model to changes in various configurations of the granular system, we further test the optimized GCN in disordered granular packings with different configurations. Interestingly, the results still show a good prediction performance, indicating the stability and effectiveness of our method.

## 2. Data Set

The data sets used in this paper are all from the data obtained from the photoelastic experiment [9]. The relevant details of the experiment are shown in Appendix A. Through the two-dimensional disk experiment, we obtained the data sets of granular systems with different disorder degrees and granular systems under different configurations. The disorder degree is expressed as the ratio of small particles (the diameter is 100 mm) to large particles (the diameter is 142 mm). In this paper, we use granular systems with disorder degrees of 1:0, 1:1, and 2.16:1, and ten different configurations of granular systems with a disorder degree of 1:1 for experiments. In order to distinguish ten different configurations of granular systems, we named these ten different configurations of granular systems conf1 to conf10. The effective particle number of each data set is in the range 1500 to 2000. With image processing techniques, we can obtain the coordinates, the diameter, and the number of neighbors of each disk in a granular system for further training.

## 3. Distance Estimation

Distance information is an important parameter in coordinate calculation. When we calculate the coordinates of an unknown object by a known object, we need to know some information between the two objects, such as distance and angle. Even if we only know the distance information, we can calculate the coordinates of unknown objects by known objects. If distance information is unknown, we can also calculate the coordinates via the following methods: (1) Some special technologies such as image processing techniques are used to obtain the coordinates of the object, but this method is limited by the material, size, and other factors of the object. (2) Estimating the distance between two objects; this method is affected by the accuracy of the algorithm.

In granular systems, when the radius of all particles is known, the distance between adjacent particles is approximately equal to the sum of the radius of two particles. If two particles are not adjacent, that is, there are one or more particles between them, then the distance between them is not equal to the sum of the distances between all particles. The distance between particles cannot be directly measured in some granular systems, due to the specificity of granular materials. However, when we want to predict the coordinates of unknown particles by known particles, we need to know the distance between two particles. Therefore, we need to predict the distance between every two particles.

In this section, we introduce an algorithm to predict the distance between every two particles. The algorithm to predict distances includes the following steps: (1) find the minimum hops between two particles and the corresponding path; (2) compute the distance estimation error and calculate the average estimation error per hop; (3) estimate the distance between nodes.

### 3.1. Building a Graph Network

Before calculating the distance between every two particles in the granular system, we need to construct the whole granular system into a graph network. The nodes of the graph network are composed of each disk of the granular system. If there is contact between two particles, edges are added to the corresponding nodes. However, such an undirected graph cannot be used to calculate the distance. For calculation convenience, we need to add weights to all edges based on the undirected graph. Therefore, we take the distance between adjacent particles as the weight of the edge to construct a weighted graph network. Figure 1 shows a graph network composed of one-hop neighbors, in which part of the data of the granular system with a disorder degree of 1 to 1 is used to construct the graph network.

### 3.2. Calculation of Shortest Path and Hop Counts

After the graph network is constructed, the next step is to predict the distance between any two non-adjacent nodes based on the graph network. Since the graph network only contains the distance between two adjacent nodes, when calculating the distance between two non-adjacent nodes, we should first find the path between them, and then use the distance information between adjacent nodes in the path to estimate it. In the granular system, there are two distance estimation methods (as shown in Figure 2). One is based on the minimum hop path, and the other is based on the shortest path. We use these two methods to calculate the distance from node 1 to node 7. According to Figure 2b, we use the minimum hop algorithm to determine that the distance is 6. According to Figure 2c, we use the shortest path algorithm to obtain a distance of 5. Obviously, the distance obtained by using the shortest path algorithm is closer to the true value, but the hop counts may be larger. In this section, in order to calculate the hop count and the shortest path between two non-adjacent nodes, we use the Dijkstra algorithm to calculate the shortest distances and hop counts between them based on the weighted graph network.

### 3.3. Estimation of the Distance between Nodes

When the shortest path and hops between two nodes are determined, we need to calculate the average error per hop, and then complete the distance estimation according to this error. First, we calculate the estimation error, according to Equation (Equation 1).
(1)$$d_{err}=d_{est}-d_{true},$$
where $d_{est}$ indicates the distance of the shortest path, equal to the sum of the distances between every two adjacent nodes in the path, and $d_{true}$ is the true distance between two nodes.

Then, we calculate the average estimation error per hop, according to Equation (Equation 2).
(2)$$E_{hop}=\frac{d_{err}}{H},$$
where $d_{err}$ is the estimation error; *H* indicates the hop counts between two nodes.

When the average estimation error of each hop is calculated, we can estimate the distance between any two nodes in the granular system with different proportions, according to Equation (Equation 3)
(3)$$d=d_{est}-E_{hop}*H,$$
where $d_{est}$ is the distance of the shortest path, $E_{hop}$ is the average estimation error per hop, and *H* indicates the hop counts in the shortest path between two nodes.

## 4. GCN Model

The undirected graph neural network converted from a granular system can be defined as G=(ν,A), where ν represents the vertex set of the nodes $\left\{\nu_{1},\nu_{2},\ldots ,\nu_{N}\right\}$, and $A\in R^{N\times N}$ is the edge set of the nodes, also called the adjacency matrix. Normally, the value of $a_{ij}\in A$ represents whether there is a connection or edge relationship between two nodes. We set $a_{ij}=0$ if there is no direct relationship or no connection between $\nu_{i}$ and $\nu_{j}$, otherwise $a_{ij}\neq 0$. In some special models, the importance of relationships between nodes is different, so the value of $a_{ij}$ can also represent the relationship coefficient between nodes *i* and *j*. The degree matrix $D\in R^{N\times N}$ is a diagonal matrix with $D=diag(d_{1},d_{2},\ldots ,d_{N})$ where $d_{i}=\sum_{j=1}^{N}a_{ij}$.

The key point of this work is how to determine whether there is a connection between two nodes; that is, how to construct a related graph network. In the context of network localization based on a GNN, one of the important factors is distance information. If the distance between the two nodes is large, then the influence between them is very small, and may not even exist. According to Section 2, distance information is related to the hop counts between particles in granular systems. Therefore, we introduce the hop counts in the shortest path between two particles, denoted by H, to determine whether or not there is an edge between two nodes. We set $a_{ij}=1$ if the minimum hop between two nodes is smaller than or equal to H; otherwise, $a_{ij}=0$, where $a_{ij}\in A_{D_{T}}$. Then, we can predict the grain-scale coordinates using the distance information.

After the graph network is determined, the following is the construction of the GCN model. In this work, we construct a GCN model with a 2-layer hidden layer [48]. In the *k*-th hidden layer, we assume that $N_{k}$ is the number of neurons in the *k*-th layer. In general, each layer of the GCN carries out three actions: feature propagation, linear transformation, and nonlinear activation, as shown in Figure 3.

In the process of feature propagation, nodes aggregate neighbor information and combine their own information to obtain a hidden representation ${\bar{H}}^{\left(k\right)}\in R^{N\times N_{k-1}}$. More precisely, in the GCN, the update process for all layers is obtained by performing the following matrix multiplication:(4)$${\bar{H}}^{\left(k\right)}=\hat{A_{D_{T}}}H^{\left(k-1\right)},$$
where $\hat{A_{D_{T}}}$ is the augmented normalized adjacency matrix with $\hat{A_{D_{T}}}={\tilde{D}}_{D_{T}}^{-\frac{1}{2}}{\tilde{A}}_{D_{T}}{\tilde{D}}_{D_{T}}^{-\frac{1}{2}}$ and ${\tilde{A}}_{D_{T}}=A_{D_{T}}+I$, ${\tilde{D}}_{D_{T}}$ is the associated degree matrix of ${\tilde{A}}_{D_{T}}$. $H^{\left(k-1\right)}\in R^{N\times N_{k-1}}$ denotes the input node representations. Intuitively, this step smooths the hidden representations locally along the edges of the graph and ultimately encourages similar predictions among locally connected nodes.

The process of linear transformation is a linear transformation of hidden representation ${\bar{H}}^{\left(k\right)}$ in the *k*-th layer; that is, ${\bar{H}}^{\left(k\right)}$ multiplied by a layer-specific trainable weight matrix $W^{\left(k\right)}$. Then, the obtained linear transformation expression is nonlinearly activated to obtain the final output of the *k*-th layer. The representation updating rule of the *k*-th layer is given by
(5)$$H^{\left(k\right)}=\phi \left({\bar{H}}^{\left(k\right)}W^{\left(k\right)}\right),$$
where ϕ(a) is a nonlinear activation function. In this work, we choose ReLU(a)=max(0,a) as a nonlinear activation function and the nonlinear activation function is applied to every element in the matrix.

When the update rules of each layer are determined, the next step is to consider the initial node representations $H^{\left(0\right)}$. In a large-scale granular system, using global distance information as the initial input will not only increase the computational complexity but also affect the prediction results. Therefore, we need to further process the original distance information. Accordingly, we construct a sparse distance matrix $\hat{X}$, which is given by
(6)$$\hat{X}=A_{D_{T}}⊙X,$$
where ⊙ denotes the Hadamard product. Consequently, $\hat{X}$ contains only distance measurements that are smaller than or equal to $D_{T}$. We take $\hat{X}$ as the initial node representations; that is, $H^{\left(0\right)}=\hat{X}$.

In this work, we need to predict the position of each particle in a granular system based on a 2-layer GCN. The estimated positions, $\hat{R}={\left[{\hat{p}}_{1},{\hat{p}}_{2},\ldots ,{\hat{p}}_{N}\right]}^{T}$, are given by
(7)$$\hat{R}=\hat{A_{D_{T}}}\phi \left(\hat{A_{D_{T}}}H^{\left(0\right)}W^{\left(1\right)}\right)W^{\left(2\right)},$$
where $W^{\left(1\right)}$ and $W^{\left(2\right)}$ are the weight matrices of the *k*-layer. We use the anchor position to train the weight matrices and optimize the weight matrices by minimizing the mean squared error (MSE), $L\left(W^{\left(1\right)},W^{\left(2\right)}\right)={∥R_{l}-\hat{R_{l}}∥}_{F}^{2}$, where $R_{l}$ and $\hat{R_{l}}$ are the true anchor positions and their estimates, respectively, and ${∥\cdot ∥}_{F}$ is the Frobenius norm of a matrix.

## 5. Results

In this section, we first discuss the positioning accuracy, which includes the root-mean-squared error and effective prediction accuracy. This part is discussed in terms of prediction experiments with different disorder degrees. Then, we focus on a granular system with a disorder ratio of 1:1, including the use of distance estimation algorithms to estimate the global distance information of granular systems with different configurations and the use of a granular system with a certain configuration as a training sample to predict granular systems with other configurations.

### 5.1. Positioning Accuracy

In this paper, we use a 2-layer GCN with 2000 neurons in each hidden layer and set the learning rate to 0.01. To verify the performance of the GCN model, we evaluate the localization accuracy. Here, the measurement of localization accuracy includes the following two aspects: (1) the averaged test root-mean-squared error (RMSE). (2) the effective prediction accuracy (EPA). The RMSE is an important indicator to measure the stability and robustness of a model, and it can also be used as the loss function of a model. It is based on the entire granular system to evaluate the model. The EPA is based on the grain scale to evaluate the prediction results of the model, which can better show the prediction effect of the model.

#### 5.1.1. The Root-Mean-Squared Error (RMSE)

The averaged test root-mean-squared error (RMSE) is given by Equation (Equation 8)
(8)$$L_{R}={∥R_{u}-{\hat{R}}_{u}∥}_{F},$$
where $R_{u}={\left[p_{N_{l}+1},p_{N_{l}+2},\ldots ,p_{N}\right]}^{T}$ denotes the actual test positions, ${\hat{R}}_{u}=[{\hat{p}}_{N_{l}+1},{\hat{p}}_{N_{l}+2},\ldots$, ${\hat{p}}_{N}{]}^{T}$ denotes the predictions of test positions, and ${∥\cdot ∥}_{F}$ is the Frobenius norm of a matrix.

Figure 4 depicts the relationship between the hop counts and root-mean-squared error (RMSE) of a granular system with three disorder degrees of 1:0, 1:1, and 2.16:1. When H is too small (H < 3), the value of RMSE seems to be small, but the prediction effect is not ideal. This is because there are not sufficient edges in the graph network when the hop count is too small, resulting in incomplete features extracted by the GCN. In this case, we can appropriately increase the number of anchor nodes to improve the prediction accuracy. When H ∈ (3, 10], from the RMSE results, RMSE is small, and the GCN seems to show good and stable performance. With the increases in hop counts, in H ∈ (10, 23], some invalid edges are added, which interferes with the model prediction, degrades the model performance, and makes the RMSE larger than the value in H ∈ (3, 10]. When H > 23, we can see that the value of RMSE increases rapidly, the prediction effect of the model is not good, and the model becomes extremely unstable. This is because when the number of node hops is too large, the sharp increase in the number of edges will lead to over-smoothing of the model (see Appendix B).

#### 5.1.2. The Effective Prediction Accuracy (EPA)

Next, we analyze the effective prediction accuracy. In order to better evaluate the performance of the GCN model, we introduce a threshold based on the root-mean-squared error to classify the prediction results. When the error of the prediction result of a point is less than the threshold, the prediction effect is better. Therefore, we believe that the prediction of this point is an effective prediction. Conversely, it is an invalid prediction; that is, the model performs too poorly at this point. We call the ratio of the number of effective predictions to the total number of nodes the effective prediction accuracy (EPA) and use it to evaluate the model.

In this paper, in order to better observe the performance of the model, we choose the threshold of 0.08; the results are shown in Figure 5. From Figure 5, we can see that when H < 3, the effective prediction accuracy (EPA) is small. When H is too small there will be no sufficient edges in the graph. When H is 3 or 4, the EPA exceeds 90%, indicating good model performance. When H ∈ (4, 12], the value of EPA decreases rapidly. The reason for such poor performance is that when some invalid edges are added to the graph network, the prediction results have interfered, which increases the coordinate error of some nodes. When H ∈ (12, 22], the value of EPA gradually increases. This is because as the number of edges increases sharply, the model begins to exhibit an over-smoothing phenomenon, and the prediction results gradually shrink inward, which reduces the error of some points. When H > 22, the value of EPA decreases rapidly and finally tends to be stable. This is because the over-smoothing phenomenon causes the model to shrink sharply and eventually shrink to a certain point.

In this paper, the accuracy, stability, and generalization ability of the GCN model in granular system coordinate prediction are tested in several granular systems with different mixing ratios. According to the analysis of RMSE and EPA, we can conclude that when the value of H is 3 or 4, the model performance is the best and the prediction accuracy is higher. Therefore, considering the influence of surrounding neighbors on node coordinate prediction and the size of the model training set, we set the hop counts(H) to 3 and the anchor node to 60% of the total number of nodes for the experiment. The experimental results for 1:0, 1:1, and 2.16:1 ratios are shown in Figure 6. The RMSE of the granular system with the ratio of 1:0 is 0.0302, and the EPA is 95.78%. The RMSE of the granular system with a ratio of 1:1 is 0.0357, and the EPA is 93.1%. The RMSE of the granular system with a ratio of 2.16:1 is 0.0417, and the EPA is 85.6%. The results for three granular systems show that the GCN model is robust.

### 5.2. Prediction of 1:1 Granular System

Next, we focus on the granular system with a disorder degree of 1:1. In order to explore the prediction performance of the GCN model for granular systems with different structures under the same disorder degree, we constructed ten different configurations of 1:1 particle systems. A detailed description of these ten different particle systems is shown in Appendix C. Firstly, we select the 1:1 granular system used in Section 5.1 as a training sample and obtain the average error of each hop according to Section 3. We then estimate the global distance information of ten different configurations of the 1:1 granular system via Equation (Equation 3). Finally, we use the estimated global distance information as the input of the GCN model to predict the coordinates of particles in the corresponding granular system. The RMSE curves of the prediction results of different configurations by the GCN are basically the same, and the EPA curve is the same. Figure 7 shows the prediction results for different configurations of granular systems with a disorder degree of 1:1 when the jump number is 3 or 4.

Figure 8 shows the prediction results for granular systems with two different configurations (conf2 and conf4). The hop count technique (H) of the two granular systems is set to 3, and the size of the training set is 1000. The RMSE value of the granular system with the configuration of conf2 is 0.0375, and the EPA value is 91.79%. The RMSE value of the granular system with the configuration of conf4 is 0.0350, and the EPA value is 93.05%.

Next, we study one of the granular systems as the training sample to train the model and predict other granular systems under the same disorder degree. When using one system to predict another system, it is first necessary to establish a connection between the two systems. This is because the GCN model uses global distance information as a feature for prediction. If the two systems do not have a distance relationship, using the GCN model for prediction will not extract any features about distance information, which will lead to a large deviation in the prediction results. The method used in this paper is to integrate all the nodes in the two systems into one system in the same coordinate system, and then calculate the global distance information of the system as the input of the GCN model. We then use one of the systems as the training sample to predict the relevant coordinate information of the other system. In this paper, we select the 1:1 granular system used in Section 5.1 (also called conf) as the training set to predict the other ten granular systems with different configurations. Figure 9 shows that the granular system of conf is used as a training sample to predict the RMSE values of other configurations of granular systems. We select 1000 node information points of the system as the training set and the remaining node information as the validation set to train the GCN model.

Figure 10 shows the predicted results for conf3 and conf8 when H = 3. The granular system of conf is used as a training sample to train the GCN model. The size of the training set is 1000, where the red points denote the real coordinate diagram of the granular system, and the blue points denote the prediction diagram. The RMSE value of the granular system with the configuration of conf3 is 0.0319, and the RMSE value of the granular system with the configuration of conf8 is 0.0325. Other results from using a granular system of conf as a training sample for prediction are shown in Appendix C.

## 6. Conclusions

In this paper, a GCN model was introduced to predict the coordinates of particles in a granular system according to the distance information of the granular system. Moreover, in the case of the same disorder degree, we used the known granular system as a sample, and the distance information of other granular systems was estimated using the distance estimation algorithm. We used granular systems with different disorder degrees to conduct experiments. The results show that the GCN model can effectively predict the particle coordinates of the granular system by extracting the distance characteristics between particles. At the same time, when the hop threshold is selected reasonably, the GCN model has strong stability. We also used a granular system with different configurations in the same disorder degree to conduct experiments. The results show that the model can predict the coordinates of the granular system well.

The key points in respect of coordinate prediction in this paper are the following: (1) Accurate prediction of the distance between non-adjacent particles. (2) Hop threshold selection. From the experimental results, the hop count has a great influence on the prediction results. When the hop threshold is set too low, the extracted features will be insufficient and the prediction results will be extremely inaccurate. When the hop threshold is set too high, this will lead to too many invalid edges added to the model, resulting in a smooth transition model, and the final prediction results will converge to one point.

Although the GCN model used in this paper can quickly realize the positioning of particles, we still have to measure the coordinates of some particles and the distance between every two particles in the unknown granular system. It is better to randomly select particles for measuring coordinates so that the GCN model can extract more comprehensive features. Under appropriate conditions, the GCN model in this paper can be extended to large-scale granular systems to achieve particle coordinate prediction, and the model can even be used to achieve coordinate prediction for three-dimensional granular systems.

## Figures and Tables

**Figure 1 micromachines-14-00714-f001:**
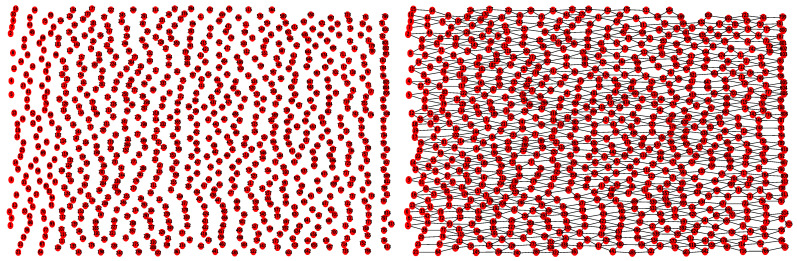
Construction of graph network. (**Left**) Partial particles of a particle system with a disorder of 1 to 1. (**Right**) Graph network constructed by one-hop neighbors of particles in left.

**Figure 2 micromachines-14-00714-f002:**
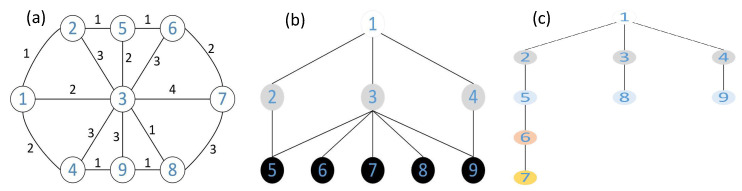
Distance estimation methods. (**a**) graph network; (**b**) minimum hop path algorithm; (**c**) shortest path algorithm.

**Figure 3 micromachines-14-00714-f003:**
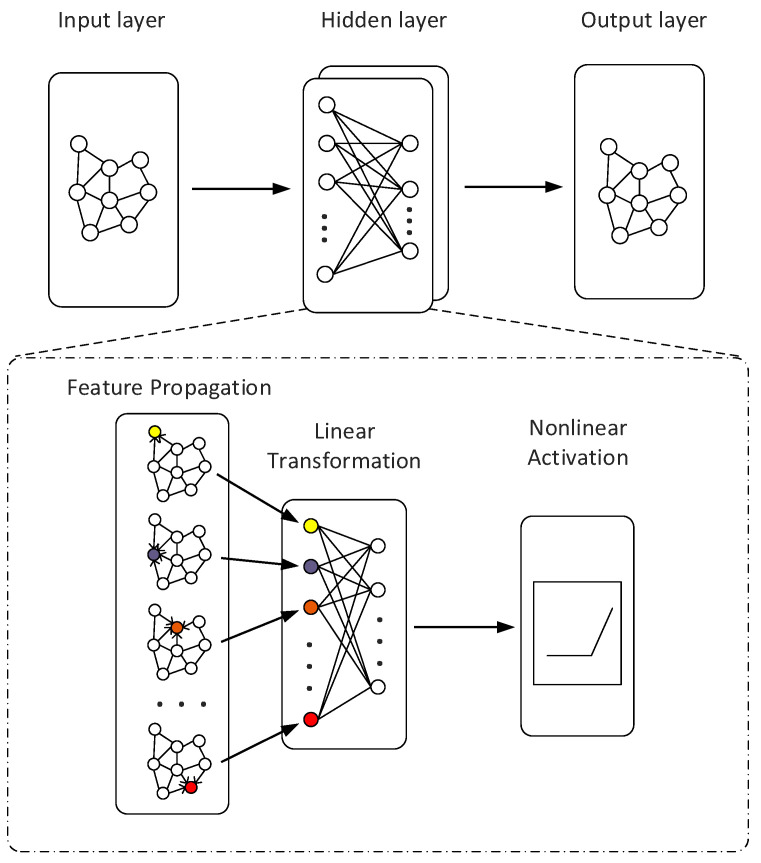
The principle of the 2-layer GCN. Diagram of GCN updating rule for hidden layer in the virtual box.

**Figure 4 micromachines-14-00714-f004:**
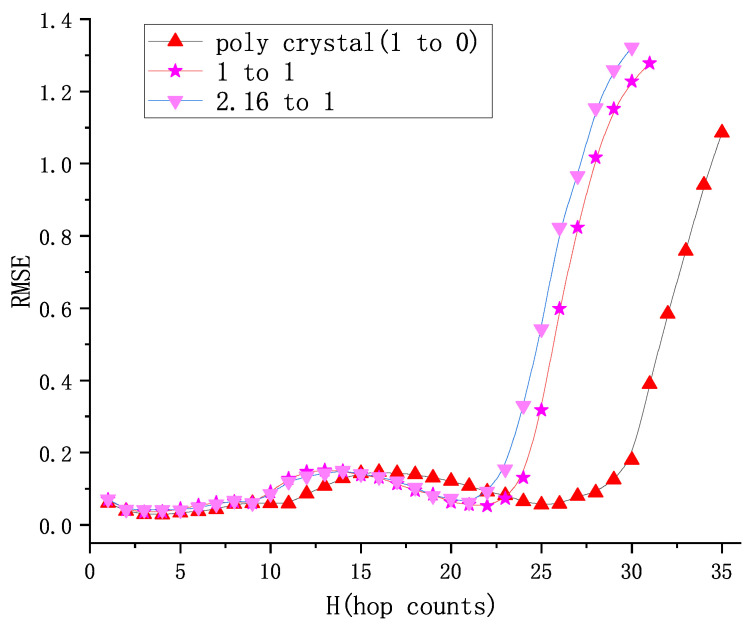
Root-mean-squared error (RMSE) versus hop counts (H). The number of anchor nodes in all systems is about 60% of the total number of nodes.

**Figure 5 micromachines-14-00714-f005:**
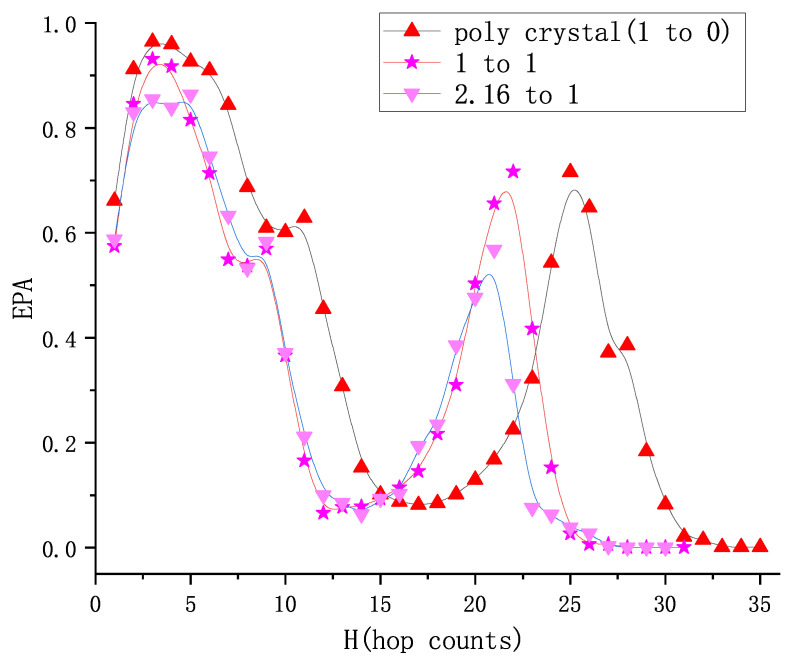
Effective prediction accuracy (EPA) versus hop counts (H). The number of anchor nodes in all systems is about 60% of the total number of nodes.

**Figure 6 micromachines-14-00714-f006:**
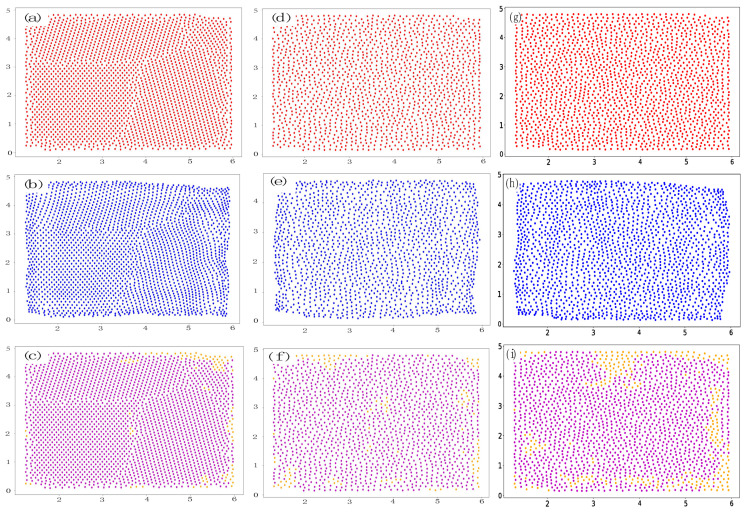
Granular system coordinate diagram, coordinate prediction diagram, and comparison diagram. The red particle map is the original coordinate map of the granular system, the blue particle map is the prediction map, the magenta particles are the effective prediction particles, and the orange particles are the ineffective prediction particles. Moreover, (**a**–**c**) show the 1:0 granular system where the RMSE is 0.0302 and the EPA is 95.78%; (**d**–**f**) show the 1:1 granular system where the RMSE is 0.0357 and the EPA is 93.1%; and (**g**–**i**) show the 2.16:1 granular system where the RMSE is 0.0417 and the EPA is 85.6%.

**Figure 7 micromachines-14-00714-f007:**
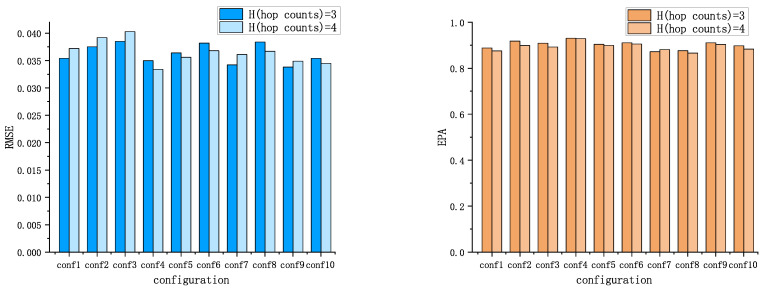
The prediction results for granular systems with different configurations of the disorder degree of 1:1. (**Left**) Root-mean-squared error (RMSE) versus configuration. (**Right**) Effective prediction accuracy (EPA) versus configuration.

**Figure 8 micromachines-14-00714-f008:**
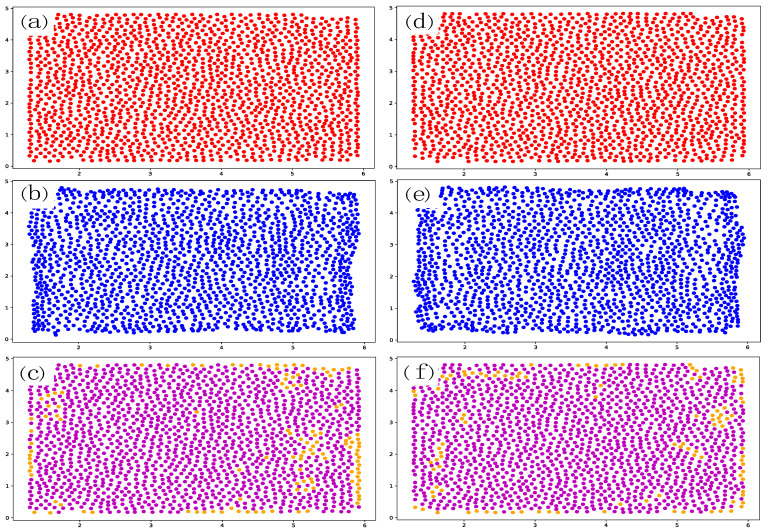
The prediction results for granular systems with different configurations of the disorder degree of 1:1. The red particle map is the original coordinate map of the granular system, the blue particle map is the prediction map, the magenta particles are the effective prediction particles, and the orange particles are the ineffective prediction particles. Moreover, (**a**–**c**) show a granular system of conf2 where the RMSE is 0.0375 and the EPA is 91.79%; (**d**–**f**) show a granular system of conf4 where the RMSE is 0.0350 and the EPA is 93.05%.

**Figure 9 micromachines-14-00714-f009:**
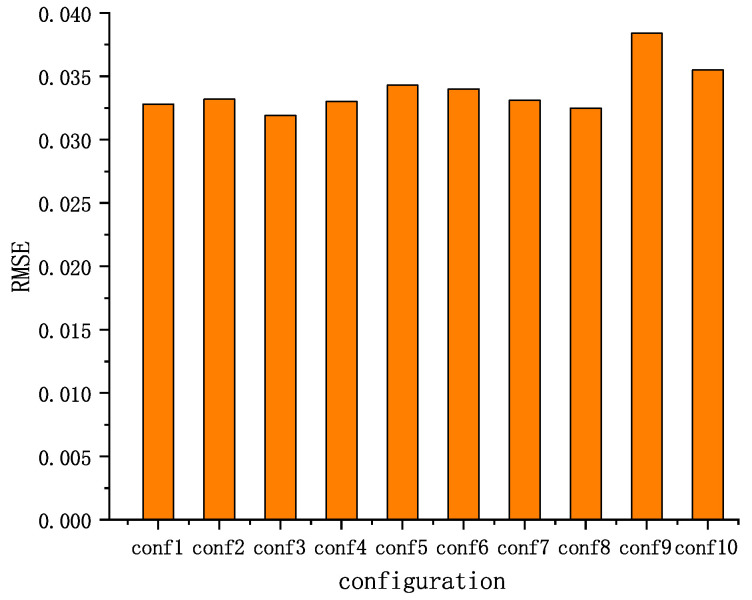
Root-mean-squared error (RMSE) versus configurations.

**Figure 10 micromachines-14-00714-f010:**
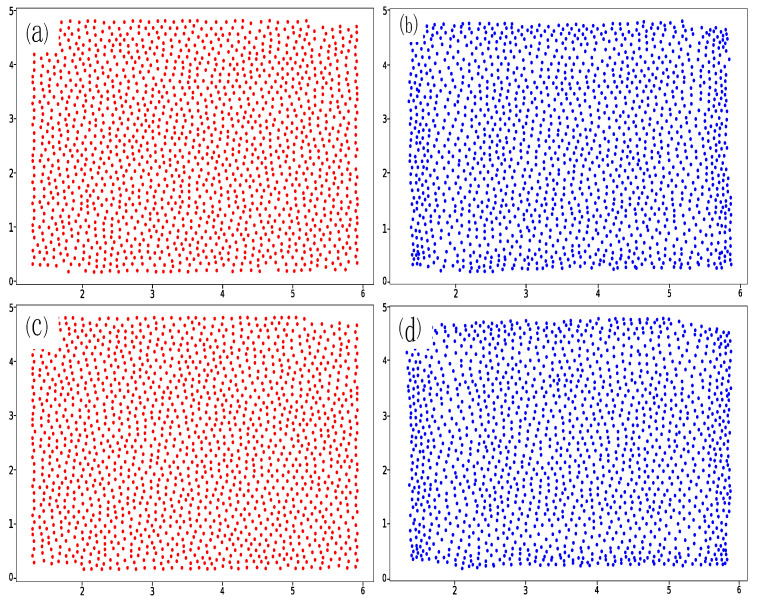
Granular system coordinate diagram and coordinate prediction diagram. (**a**,**b**) are the granular system coordinate diagram and the coordinate prediction diagram of conf3. The RMSE of conf3 is 0.0319. (**c**,**d**) are the granular system coordinate diagram and the coordinate prediction diagram of conf8. The RMSE of conf8 is 0.0325.

## Data Availability

All data can be obtained from the authors under reasonable request.

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
