# Peer review of "Estimation of Particle Location in Granular Materials Based on Graph Neural Networks"

_micromachines, 2023, doi:10.3390/mi14040714_

Round 1
Reviewer 1 Report
The authors propose to use neural network approaches to predict the behavior of two-dimensional granular systems by analyzing the distances between particles. The problems of improving the efficiency of network training, the stability of its operation, and the reliability of the results are discussed. The work may be accepted for publication.
Author Response
The authors propose to use neural network approaches to predict the behavior of two-dimensional granular systems by analyzing the distances between particles. The problems of improving the efficiency of network training, the stability of its operation, and the reliability of the results are discussed. The work may be accepted for publication.
We are grateful to the reviewer 1 for recommending our work to Micromachines now, thank you so much for your time and effort in reviewing our manuscripts.
Reviewer 2 Report
This paper proposes a particle position prediction model based on the graph neural network, which constructs a weighted graph of adjacent particles and calculates the distance of non-adjacent particles by graph theory algorithm, then predicts the position of each particle based on graph convolutional network, and finally verifies the validity of model prediction by experiments. There are some concerns as follows:
(1) The author proposes to use a novel graph neural network, but GCN is a classic graph neural network algorithm. The author should introduce its novelty more in detail in the paper.
(2) There are many references in the introduction, some of which are quite old. It is recommended to update the literature to the latest. Only those papers that materially support or extend discussions of your work should be cited.
(3) The image on the right side of Figure 1 is blurry, and vector graphics are recommended for displaying it.
(4) There are citation invalids in section 3.2, and please check the manuscript carefully before submitting it.
(5) In the paper, the authors mentioned that the model has strong robustness. However, the performance of the downstream task can only prove the effectiveness of the method, instead of its robustness.
(6) It is recommended to add the specific results of the experiments to Figure 6, Figure 8, and Figure 10 so that readers can view them more intuitively.
(7) There is no numerical description of experimental results in Appendix C.
(8) I suggest designing comparative experiments with other models to evaluate the performance and accuracy of the innovation more fully.
(9) There are some grammar errors in this paper.
Author Response
This paper proposes a particle position prediction model based on the graph neural network, which constructs a weighted graph of adjacent particles and calculates the distance of non-adjacent particles by graph theory algorithm, then predicts the position of each particle based on graph convolutional network, and finally verifies the validity of model prediction by experiments.
We appreciate the time and efforts of the reviewer 2 in reviewing our manuscript. With the reviewers' suggestions, the quality of our revised manuscript has been improved a lot.
The author proposes to use a novel graph neural network, but GCN is a classic graph neural network algorithm. The author should introduce its novelty more in detail in the paper.
We regret that we ignore the detailed description of the improvements to the GCN model, thank you for your careful review and friendly reminder. Compared with the traditional GCN model, the GCN used in this paper increases the hop threshold and the hidden representation of nodes. Of course, this does not mean that the model is a novel GCN model. Therefore, we corrected this statement in the new revision and explained it in line 48, page 2.
Action: We use ‘an improved graph neural network’ to replace ‘a novel graph neural network’ and explain the improvement of the model in line 48, page 2.
There are many references in the introduction, some of which are quite old. It is recommended to update the literature to the latest. Only those papers that materially support or extend discussions of your work should be cited.
Yes, some of the methods mentioned in this article have been updated in recent years, the old literature on these methods may not be supported or extended discussions of our work. Therefore, we carefully review the references and make some changes to the references to ensure that all references cited are materially support or extend discussions of our work.
Action: Now references have been updated. Such as Buchenau, et al., Physical Review Letters 1984, 53, 2316-2319. and Blanshard, et al., Carbohydrate Polymers 1984, 4, 427-442. are replaced by Biswas, et al., Powdwe Technology 2021, 378, 680-684.; Watanabe, et al., Physical Review Letters 2008, 100, 158002. is replaced by Tapia-Ignacio, et al., Physical Review E 2016, 94, 062902. and so on.
The image on the right side of Figure 1 is blurry, and vector graphics are recommended for displaying it.
We regret that we overlooked this problem, thank you for your careful review and friendly reminder.
Action: Now it has been replaced by vector graphics.
There are citation invalids in section 3.2, and please check the manuscript carefully before submitting it.
We feel so apologized that our negligence led to citation invalids in section 3.2, and we have carefully reviewed the new revised version to ensure that similar problems do not occur again.
Action: Now it has been corrected.
In the paper, the authors mentioned that the model has strong robustness. However, the performance of the downstream task can only prove the effectiveness of the method, instead of its robustness.
There are different opinions on the definition of robustness in the field of machine learning, but there is a consensus that if a machine learning model can still guarantee the stability of the model under disturbance, it can be considered as robust. Although the experimental results of ten different configurations of granular systems may be more to prove the effectiveness of the model, we also predicted the positions of three granular systems with different degrees of disorder. The RMSE of the 1:0, 1:1, and 2.16:1 is 0.0302, 0.0357, and 0.0417, respectively. And the EPA of the the 1:0, 1:1, and 2.16:1 is 95.78%, 93.1, and 85.6, respectively. This also proves the robustness of the model. In the subsequent research, we will extened the GCN model to the granular systems with different volume fraction, different interaction and more complex systems such as large-scale granular systems and three-dimensional granular systems.
Action: The ‘robustness’ has been replaced by ‘effectiveness’ in line 7, page 1, line 112, page 3, and line 444, page 14.
It is recommended to add the specific results of the experiments to Figure 6, Figure 8, and Figure 10 so that readers can view them more intuitively.
We appreciate your careful review of our manuscript. Since we have a detailed description of the experimental results in the paper, we did not add the specific results of the experiment to the figure. But the specific results of the experiments should be added to figure to ensure readers can view them more intuitively. Therefore, we have carefully checked all the pictures and added specific experimental results.
Action: The specific results are added to Figure 6, Figure 8, Figure 10 and Figure A1.
There is no numerical description of experimental results in Appendix C.
Firstly, the GCN model training set and test set are from a two-dimensional photoelastic experiment, rather than a numerical simulation. There is a detailed description of the experiment in Appendix A. In order to obtain ten different configurations of 1:1 granular system, we take out the particles after each experiment and place them randomly, so that the ten particle systems have completely different structures. And the detailed description of experiment to the 1:1 granular systems with differen configurations in line 423, page 14. Moreover, the prediction results of the GCN model are described in detail in Appendix C, see line 438, page 14.
Action: The experimental methods of different configurations and the prediction results of GCN model are shown in Appendix C.
I suggest designing comparative experiments with other models to evaluate the performance and accuracy of the innovation more fully.
In order to evaluate the performance and accuracy of innovation more comprehensively, we should indeed design comparative experiments with other models. However, to our knowledge, the usual methods to extract the particle locations are either referred to some real experimental methods (e.g. neutron scattering, x-ray, electron microscopy or other imaging systems), or referred to some numerical simulation methods (Lammps or DEM) , and presently, there is few report related to the prediction of particle locations using machine learning methods in the granular field. Moreover, most of the existing machine learning models needs to be input some complex handcrafted features for training, and we have to define and precompute them properly combined with the traditional physical intuition, which is not easy for us within a short time, but recently we are indeed exploring other machine learning models for coordinates prediction in granular materials as you suggest. Thank you so much for your kind suggestion.
There are some grammar errors in this paper.
We appreciate your careful review of our manuscript. And we regret that tThere are some grammar errors in this paper, thank you for your careful review and friendly reminder. We carefully reviewed the manuscript and corrected these grammar errors in our new revised manuscript.
Action: Now the grammar errors in the paper have been corrected.
Reviewer 3 Report
The authors introduce a GCN model for predicting particle coordinates in granular systems based on distance information, and develop a distance estimation algorithm to estimate the distance information of granular systems with different configurations and disorder degrees. The results demonstrate the robustness and effectiveness of the GCN model for predicting particle coordinates in granular systems with different configurations and disorder degrees. Generally, the GCN application in predicting particle coordinates in granular systems is valuable. However, several minor revisions are needed. First, the article does not provide a comparison of the proposed GCN model with existing models or methods for predicting particle coordinates in granular systems. Second, the article does not provide a discussion of potential limitations or challenges of applying the proposed model to real-world granular systems.
Author Response
The authors introduce a GCN model for predicting particle coordinates in granular systems based on distance information, and develop a distance estimation algorithm to estimate the distance information of granular systems with different configurations and disorder degrees. The results demonstrate the robustness and effectiveness of the GCN model for predicting particle coordinates in granular systems with different configurations and disorder degrees. Generally, the GCN application in predicting particle coordinates in granular systems is valuable.
We are very grateful to reviewer 3 for the advice and affirmation of our work, and thank you very much for your time and efforts to review our manuscripts. With the reviewers' suggestions, the quality of our revised manuscript has been improved a lot.
The article does not provide a comparison of the proposed GCN model with existing models or methods for predicting particle coordinates in granular systems.
In order to evaluate the performance and accuracy of innovation more comprehensively, we should indeed design comparative experiments with other models. However, to our knowledge, the usual methods to extract the particle locations are either referred to some real experimental methods (e.g. neutron scattering, x-ray, electron microscopy or other imaging systems), or referred to some numerical simulation methods (Lammps or DEM) , and presently, there is few report related to the prediction of particle locations using machine learning methods in the granular field. Moreover, most of the existing machine learning models needs to be input some complex handcrafted features for training, and we have to define and precompute them properly combined with the traditional physical intuition, which is not easy for us within a short time, but recently we are indeed exploring other machine learning models for coordinates prediction in granular materials as you suggest. Thank you so much for your kind suggestion.
The article does not provide a discussion of potential limitations or challenges of applying the proposed model to real-world granular systems.
Yes, indeed. Thank you for your careful review and friendly reminder. We have discussed the potential limitations of the model in our new revised manuscript. Moreover, we also discuss the extended application of the model.
Action: We discuss this problem in the conclusion section of the revised manuscript, see line 377, page 13.
Round 2
Reviewer 2 Report
The quality of the revised manuscript has been greatly improved and can be accepted for publication.